# StructGen: Leveraging Structured EHR Prompts and Biomedical BERTs for Chest X-ray Synthesis

Suchit Patel*†, Karandeep Singh Sodhi*, Manik Gupta*, Mei-Tai Chu †

* Department of CSIS, BITS-Pilani Hyderabad Campus, India, 500078;
E-mail: {p20230024, f20221383, manik}@hyderabad.bits-pilani.ac.in
† La Trobe Business School, La Trobe University, Melbourne, Australia, VIC 3086; E-mail: M.Chu@latrobe.edu.au

*Abstract*—The generation of synthetic chest X-rays from textual clinical data has shown significant promise for augmenting medical datasets and supporting downstream diagnostic tasks. This study extends the RoentGen framework, a latent diffusion-based image generator, by systematically evaluating the influence of structured Electronic Health Record (EHR) derived prompt types and domain-specific Bidirectional Encoder Representations from Transformers (BERT)-based language models on image quality and semantic fidelity. We propose four prompt strategies derived from structured EHR fields: Detailed, Disease, Demographic, and Device, and examine their effect on synthesized image realism and alignment. Additionally, we compared ten biomedical encoders, including ClinicalBERT, BioBERT, PubMedBERT, and others, across multiple visual-semantic metrics such as Structural Similarity Index Measure (SSIM), Peak Signal-to-Noise Ratio (PSNR), Learned Perceptual Image Patch Similarity (LPIPS), Contrastive Language Image Pretraining Score (CLIPScore), and Fréchet Inception Distance with X-Radiology Vision features (FID-XRV). Our findings highlight that both the content of the prompt and the choice of encoder substantially impact the quality and interpretability of generated images. Notably, BioBERT paired with disease-centric prompts consistently yields superior results. This work provides valuable insights for improving conditional medical image generation, particularly in settings with limited narrative text.

*Index Terms*—RoentGen, medical image synthesis, CXR generation, clinical text conditioning, biomedical language models, Prompt engineering.

## I. INTRODUCTION

Chest X-rays (CXRs) are among the most widely used diagnostic tools in medicine, essential for identifying thoracic conditions such as pneumonia, pleural effusion, and cardiomegaly [1]. However, developing robust machine learning models for CXR interpretation is hindered by data imbalance, annotation scarcity, and privacy constraints. Synthetic image generation has emerged as a promising solution for data augmentation, simulation-based training, and explainable AI [2]. In particular, generating medical images from textual inputs such as radiology reports and clinical notes addresses the need for large, annotated datasets. Yet, traditional generative models often struggle to capture the complex semantics of medical data, necessitating domain-specific approaches.

Recent advances in generative modeling, especially through latent diffusion models (LDMs), have enabled controllable and high-fidelity medical image synthesis conditioned on textual descriptions. RoentGen [3] is a recent vision-language framework that uses a frozen CLIP text encoder [4] to transform radiology reports into embeddings that guide chest X-ray generation. By fine-tuning on paired CXR images and corresponding textual reports, RoentGen demonstrates the capability to produce diverse and clinically plausible images, offering a valuable tool for data augmentation and educational purposes. However, its dependence on free-text reports and a single encoder architecture restricts flexibility and raises questions about generalizability in real-world scenarios. In many clinical contexts, structured Electronic Health Record (EHR) data, such as patient age, gender, race, diagnostic codes, and device usage, are more readily available than detailed radiology reports and can offer an alternative input modality for image generation. Effectively leveraging this structured data for image synthesis requires innovative prompt engineering strategies that can translate discrete clinical attributes into meaningful textual prompts and selecting appropriate biomedical text encoders for conditioning remains an open research problem.

This gap motivates our work to evaluate how the choice of biomedical language models affects conditioning in latent diffusion-based CXR generation. This study investigates how prompt engineering from structured EHR fields and the choice of biomedical language models like Bidirectional Encoder Representations from Transformers (BERT) [5], Biomedical Bidirectional Encoder Representations from Transformers (BioBERT) [6] and PubMedBERT [7], which have shown superior performance in medical natural language processing (NLP) tasks, influence the quality and clinical validity of generated CXR images in comparison to the RoentGen base model. We selected a diverse set of domain-specific text encoders based on their pretraining on biomedical and clinical datasets. This diversity allowed us to systematically evaluate how variations in pretraining data sources (e.g., biomedical literature, clinical notes, discharge summaries) and architectural adaptations (e.g., long-sequence handling, relational linking) influence the fidelity, realism, and semantic alignment of generated medical images [3]. These models represent the current landscape of specialized language models optimized for healthcare-related NLP tasks. In StructGen, we extend the RoentGen pipeline to incorporate diverse prompt formulations and multiple BERT-based domain-specific encoders to evaluate

This project (Project ID: 33-EOI-BITSP) is funded under the Asian Smart Cities Research Innovation Network (ASCRIN) program between BITS Pilani Hyderabad Campus (India) and La Trobe University (Australia).

their impact across various metrics to enhance the semantic and pathological realism of synthetic images. The key contributions of this work are as follows:

- A comparative analysis of ten biomedical language models as text encoders for conditioning latent diffusion models in RoentGen CXR generation.
- A systematic study of prompt engineering strategies using structured EHR data to drive clinically meaningful image synthesis.
- A comprehensive evaluation framework assessing fidelity, diversity, and diagnostic alignment of generated images under different prompt-encoder combinations.

Through comprehensive experiments and evaluations, we seek to enhance the applicability of text-conditioned image generation in clinical contexts, particularly in settings with limited access to detailed radiology reports.

## II. LITERATURE SURVEY

This section presents a comprehensive review of the current state-of-the-art in two key areas central to this study: (1) text-to-image generation with a focus on medical imaging applications, and (2) biomedical language models designed for processing clinical and domain-specific text. The goal is to contextualize our work within recent advances and identify existing gaps that motivate the need for evaluating prompt engineering strategies and encoder selection in medical image synthesis.

### A. Text-to-Image Generation in Medical Imaging

Generative models have transformed computer vision, and their adaptation to medical imaging has led to meaningful advances in data synthesis. Early works focused on GAN-based models. For example, MedGAN [2] introduced a framework for generating multi-modal medical images using adversarial loss and perceptual regularization. In [8], the authors have used GANs for liver lesion augmentation, improving Convolutional Neural Network (CNN) classification accuracy. Text-guided generation has emerged more recently with the development of models such as AttnGAN [1] and Text to Image (T2I) model [9] in the general domain. In medical contexts, Taming Transformers [10] and Guided Language to Image Diffusion for Generation and Editing (GLIDE) [11] inspired the use of text-conditioning for image generation. Medfusion [12] adapted these concepts using medical language prompts to guide diffusion-based generation. RoentGen [3] builds upon this line of work by leveraging a latent diffusion model conditioned on embeddings derived from CLIP text encoder. While it achieves state-of-the-art performance in radiology image synthesis verified by radiologists, its reliance on radiology reports limits its adaptability to real-world clinical systems, where structured metadata is often the only available input. Moreover, its use of a single encoder restricts semantic diversity in generation, motivating our study.

### B. Biomedical Language Models for Clinical NLP

Transformer-based language models pretrained on biomedical datasets have significantly improved performance in clinical NLP tasks such as named entity recognition (NER), relation extraction (RE), and question answering (QA). Among the most prominent models are BERT [5], BioBERT [6], BioLinkBERT [13], BlueBERT [14], Clinical-Longformer [15], ClinicalBERT [16], DischargeBERT [17], Medical Robustly Optimized BERT Approach (MedRoBERTa) [18], PubMedBERT [7], RedBERT [19], and Self-Alignment Pretraining for Biomedical Entity Representation (SapBERT) [20]. Table I discusses the different hyperparameters and characteristics of these models. These models refine representation learning by incorporating ontology alignment, scientific vocabulary, and link prediction into training. While these models have demonstrated success in traditional NLP tasks, their comparative performance in vision-language tasks, especially in guiding image synthesis, remains underexplored.

## III. METHODOLOGY

This section outlines the methodological framework adopted in this study, encompassing all critical components required for analyzing the effect of prompt design and language model selection on chest X-ray generation. The pipeline is systematically divided into five key stages: Dataset Description, Prompt Generation Strategies, Biomedical Language Models, Image Generation Pipeline, and Evaluation Metrics. Together, these components form a modular, yet cohesive pipeline designed to evaluate how structured clinical information and encoder selection impact the semantic controllability and diagnostic fidelity of synthetic chest X-ray generation.

### A. Dataset Description

This study leverages structured EHR data derived from the MIMIC-IV database [21] and MIMIC-CXR [22], a large-scale, publicly available dataset containing comprehensive clinical information for patients admitted to the intensive care units (ICUs) and their corresponding chest x-rays of the Beth Israel Deaconess Medical Center. MIMIC-IV comprises over 27 interlinked tables capturing demographic, diagnostic, procedural, and laboratory data. For the purposes of this study, we construct a curated dataset focused on metadata relevant to chest X-ray generation. MIMIC-CXR contains $377,110$ radiographs and $227,827$ accompanying reports, each linked to unique MIMIC-IV identifiers. The final dataset includes structured fields extracted and merged across the relevant MIMIC-IV tables, along with their corresponding CXR images. The selected features include both demographic attributes and clinical indicators:

- `disease`: The target pathology or diagnosis label used as a proxy for clinical findings.
- `support_devices`: Binary indicator denoting the presence of medical devices (e.g., ventilators, tubes).
- `gender`: Boolean field representing patient gender (Male: 0 and Female: 1).

| Model | Architecture | Training Corpus | Max Input Length | Domain Focus | Special Features |
|---|---|---|---|---|---|
| BERT [5] | BERT-base | Wikipedia + BookCorpus | 512 | General domain | Foundational transformer model; serves as a baseline for domain adaptation |
| BioBERT [6] | BERT-base | PubMed abstracts, PMC full-text articles | 512 | Biomedical literature | Early domain-specific BERT variant; widely used in NER and QA |
| BioLinkBERT [13] | BERT-base | PubMed + document link graphs | 512 | Biomedical literature with citations | Exploits inter-document links using link-aware attention |
| BlueBERT [14] | BERT-base | PubMed + MIMIC-III clinical notes | 512 | Biomedical and Clinical | Hybrid corpus for cross-domain NLP |
| Clinical-Longformer [15] | Longformer | Clinical notes (e.g., MIMIC-III) | 4096+ | Clinical narratives | Sparse attention for long documents (EHRs, summaries) |
| ClinicalBERT [16] | BERT-base | MIMIC-III clinical notes | 512 | Clinical (EHR) | Trained on real-world hospital text; supports predictive analytics |
| DischargeBERT [17] | BERT-base | MIMIC-III discharge summaries | 512 | Discharge documents | Specialized for discharge notes and patient outcomes |
| MedRoBERTa [18] | RoBERTa-base | PubMed + PMC | 512 | Biomedical | Robust optimization, dynamic masking, better generalization |
| PubMedBERT [7] | BERT-base (trained from scratch) | PubMed abstracts only | 512 | Biomedical terminology | Domain-specific tokenizer and pre-training corpus |
| RadBERT [19] | BERT-base | Radiology reports (e.g., MIMIC-CXR) | 512 | Radiology / Imaging text | Tailored for radiological NLP tasks |
| SapBERT [20] | BERT-base | UMLS ontology with PubMed/MIMIC | 512 | Biomedical entity linking | Aligns synonymic entities in UMLS using self-alignment training |

- `race`: Patient's self-identified racial or ethnic background.
- `anchor_age`: Patient's age at the time of admission or examination.

This structured representation is employed as a substitute for free-text radiology reports to simulate realistic clinical scenarios, where only metadata is available. The intention is to evaluate how effectively prompt generation techniques can translate such tabular data into informative conditioning prompt inputs for image generation.

### B. Prompt Generation Strategies

To evaluate how different types of EHR-derived metadata influence image generation, we design four distinct prompt generation strategies that convert structured EHR clinical data into descriptive natural language prompts. These strategies aim to cover varying levels of clinical detail and linguistic naturalness.

*1) Full Detailed Prompt (Rule-based):* This prompt generation method combines all available metadata of radiological conditions, device information, and patient demographics using a deterministic, rule-based template. This composite prompt provides the most complete structure to natural mapping. This level of detail mirrors realistic radiology report summaries and allows evaluation of generation under maximum clinical grounding. *Example: "Chest X-ray showing pneumonia with support devices present. Patient details: Hispanic ethnicity, age 58, Gender Male."* for EHR Data: Disease = Pneumonia; Device = Clips; Ethnicity = Hispanic; Age = 58; Gender = Male

*2) Disease Prompt:* This strategy involves generating a concise prompt that mentions only the primary radiological finding or condition extracted from the EHR or radiology report. This minimal representation serves as a baseline to evaluate the effect of richer contextual information in other strategies. This prompt is direct and medically focused, making it ideal for assessing whether the model can synthesize disease-relevant features in isolation. *Example: "Chest X-ray showing pneumonia."* for EHR Data: Disease = Pneumonia

*3) Demographics Prompt (Demo):* This strategy enriches the prompt with patient demographic attributes such as age, gender, and ethnicity, which can influence disease presentation and imaging appearance. Incorporating demographics enables analysis of whether generative models reflect realistic population-level variation or introduce demographic bias. *Example: "Chest X-ray showing Patient details: Asian ethnicity, age 62, Gender Female."* for EHR Data: Ethnicity = Asian; Age = 62; Gender = Female

*4) Device Prompt:* In this strategy, we augment the prompt with information about any medical devices present in the image, such as chest tubes, catheters, or ventilators. Such additions provide richer clinical context that may guide the model in generating more medically accurate visualizations. Including device information is crucial, especially for ICU or post-operative patients, where support devices are prevalent. *Example: "Chest X-ray with support devices present."* for EHR Data: Device = Surgical clips

Each prompt generation strategy reflects different priorities in EHR-to-text conversion, ranging from richly to minimal detailed inputs, and helps assess how semantic granularity affects image fidelity, diagnostic realism, and bias sensitivity. These strategies also enable systematic ablation studies to analyze which metadata types most contribute to generating clinically accurate images.

## C. Biomedical Language Models

To generate clinically meaningful images, we evaluated ten biomedical BERT-based language models as text encoders. These models, varying in pretraining data and representational focus, produce contextual embeddings that condition the RoentGen diffusion pipeline. Each is optimized for biomedical or clinical NLP tasks, allowing us to assess how different textual semantics impact image synthesis.

**BERT** is a general-purpose transformer model that captures deep bidirectional context by jointly attending to left and right sequences [5]. Despite not being domain-specific, it provides strong performance across various NLP tasks due to its robust pretraining. **BioBERT** extends BERT by further pretraining on biomedical corpora [6], enabling improved handling of biomedical terminology and outperforming baselines in NER, RE, and QA. **BioLinkBERT** enhances BERT with document link-aware attention to capture inter-document relationships like citations [13], benefiting tasks such as cross-document QA and scientific classification. **BlueBERT** is pretrained on both PubMed abstracts and MIMIC-III clinical notes [14], enabling cross-domain adaptability between biomedical literature and clinical narratives. **Clinical-Longformer** adapts Longformer's sparse attention to handle long clinical texts efficiently (up to $4,096+$ tokens) [15], making it suitable for lengthy medical documents. **ClinicalBERT** is BERT further pretrained on MIMIC-III clinical notes [16], enhancing its ability to interpret clinical jargon and abbreviations for decision-support tasks. **DischargeBERT** specializes Clinical-BERT for discharge summaries [17], capturing critical clinical information to support tasks like readmission prediction and outcome modeling. **MedRoBERTa** adapts RoBERTa for biomedical domains using dynamic masking and extended training [18], improving stability and performance in classification and similarity tasks. **PubMedBERT** is trained from scratch on PubMed abstracts with a domain-specific vocabulary [7], yielding superior results on tasks requiring precise biomedical language understanding. **RadBERT** is optimized for radiology reports [19], excelling in tasks like impression summarization and report classification by capturing domain-specific imaging language. **SapBERT** aligns embeddings of synonymous biomedical terms using the UMLS ontology [20], significantly enhancing performance in entity linking and concept normalization.

Each model was loaded via the HuggingFace Transformers library and used to generate token embeddings from each prompt. The embeddings were pooled and projected to match the dimensional expectations of the RoentGen conditioning input. Each model is frozen and used to generate prompt embeddings that condition the RoentGen image generation pipeline.

## D. Image Generation Pipeline

RoentGen [3] employs a latent diffusion model (LDM) conditioned on text embeddings to generate chest X-ray images from natural language prompts, as shown in Fig. 1. The pipeline consists of the following stages:

*1) Prompt Conditioning:* Each generation task begins with a prompt $p_{ij}$, where $i$ denotes the strategy (e.g., disease description, demographic information), and $j$ identifies the encoder. Prompts describe clinical attributes to be reflected in the generated image (e.g., "a chest X-ray showing signs of pneumonia").

*2) Text Embedding:* The prompt $p_{ij}$ is passed through a domain-specific text encoder, Encoder$_j$ (e.g., ClinicalBERT, BioBERT) to produce a semantic embedding $\mathbf{z}_{ij}$:

$$\mathbf{z}_{ij} = \text{Encoder}_j(p_{ij}) \qquad (1)$$

This embedding captures clinically relevant information and serves as the conditioning input for image synthesis. The choice of encoder directly impacts the semantic alignment of the output.

*3) Latent Diffusion and Image Generation:* The latent diffusion model uses $\mathbf{z}_{ij}$ to guide the denoising process in latent space, generating a synthetic chest X-ray $\hat{I}_{ij}$:

$$\hat{I}_{ij} = \text{RoentGen}(\mathbf{z}_{ij}) \qquad (2)$$

This process refines a noise vector into an image that visually aligns with the semantic content of the prompt. The final output reflects the clinical features described, enabling applications in medical education, simulation, and data augmentation.

## E. Evaluation Metrics

To assess the quality and alignment of the generated medical images with their textual prompts, we employ a set of well-established quantitative evaluation metrics. These metrics evaluate aspects such as structural similarity, perceptual similarity, semantic consistency, and realism.

*1) Structural Similarity Index Measure (SSIM):* SSIM is a perceptual metric that quantifies the similarity between two images based on luminance, contrast, and structural information [23]. Given two image patches $x$ and $y$, SSIM is computed as:

$$\text{SSIM}(x,y) = \frac{(2\mu_x\mu_y + C_1)(2\sigma_{xy} + C_2)}{(\mu_x^2 + \mu_y^2 + C_1)(\sigma_x^2 + \sigma_y^2 + C_2)} \qquad (3)$$

where $\mu_x$ and $\mu_y$ are the means of $x$ and $y$, $\sigma_x^2$ and $\sigma_y^2$ are the variances, $\sigma_{xy}$ is the covariance between $x$ and $y$, and $C_1$, $C_2$ are stabilization constants. SSIM values range from 0 to 1, with higher values indicating greater structural similarity.

*2) Peak Signal-to-Noise Ratio (PSNR):* PSNR is a widely used metric that measures the pixel-level fidelity between a generated image and its ground truth counterpart [24]. It is derived from the mean squared error (MSE) between the two images, expressed in decibels (dB). The formula for PSNR is:

$$\text{PSNR} = 10 \cdot \log_{10}\left(\frac{MAX_I^2}{\text{MSE}}\right) \qquad (4)$$

where $MAX_I$ is the maximum possible pixel value of the image (e.g., 255 for 8-bit images). Higher PSNR values indicate lower distortion and better image quality. While PSNR is sensitive to small pixel differences, it does not account for perceptual or semantic fidelity.

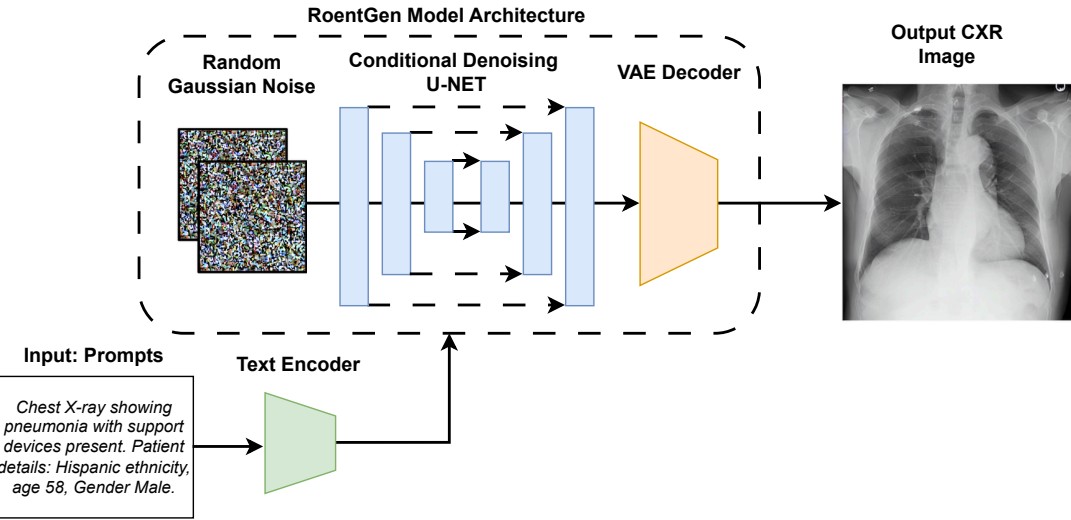

Fig. 1. Overall StructGen Architecture

*3) Learned Perceptual Image Patch Similarity (LPIPS):*
LPIPS measures perceptual similarity by computing distances between deep feature representations extracted from a pre-trained network (AlexNet in this case) [25]. Given two images $x$ and $y$, the LPIPS score is:

$$\text{LPIPS}(x,y) = \sum_l \frac{1}{H_l W_l} \sum_{h,w} \|w_l \odot (f_l^x(h,w) - f_l^y(h,w))\|_2^2 \tag{5}$$

where $f_l^x$ and $f_l^y$ are feature maps from layer $l$, $w_l$ is a learned weight, and $H_l, W_l$ are spatial dimensions of the layer. Lower LPIPS scores indicate higher perceptual similarity.

*4) CLIPScore:* CLIPScore evaluates the semantic alignment between a text prompt and a generated image using the CLIP (Contrastive Language–Image Pretraining) model [26]. It computes the cosine similarity between the image and text embeddings:

$$\text{CLIPScore}(I,T) = \cos(\phi_{\text{img}}(I), \phi_{\text{text}}(T)) \tag{6}$$

where $\phi_{\text{img}}$ and $\phi_{\text{text}}$ are CLIP image and text encoders, respectively. A higher CLIPScore indicates stronger semantic correspondence between the prompt and the image.

*5) Fréchet Inception Distance (FID-XRV):* FID measures the distance between distributions of real and generated image features, capturing both quality and diversity [27]. In the FID-XRV variant, features are extracted using a medical domain-specific encoder (e.g., CheXNet or another radiology network). The score is calculated as:

$$\text{FID}(X,Y) = \|\mu_X - \mu_Y\|_2^2 + \text{Tr}(\Sigma_X + \Sigma_Y - 2(\Sigma_X \Sigma_Y)^{1/2}) \tag{7}$$

where $(\mu_X, \Sigma_X)$ and $(\mu_Y, \Sigma_Y)$ are the mean and covariance of features from the real and generated images, respectively. Lower FID scores suggest that the generated images closely resemble real ones in distribution.

*F. Experimental Setup*

To systematically evaluate the effect of biomedical language models on medical image generation, we perform a full combinatorial analysis over 4 Prompt Strategies × 10 Text Encoders = 40 unique generation configurations. For each configuration, $N = 500$ synthetic chest X-ray images are generated and evaluated independently. All x-ray generations are conducted using the same underlying latent diffusion architecture, specifically the RoentGen model [3], with identical U-Net, scheduler, and decoder parameters across all experiments. This uniformity ensures that observed performance differences can be attributed solely to the choice of text encoder and prompt structure, rather than architectural or hyperparameter variation. To preserve experimental reproducibility, each configuration was seeded deterministically and processed on a shared computational environment. This architectural adjustment permits more accurate grounding of medical image generation in biomedical text, resulting in images that not only align with the visual appearance of X-rays, but also reflect clinical specificity embedded in the textual descriptions.

## IV. RESULT AND DISCUSSION

This section presents a detailed evaluation of model performance based on four quantitative metrics: SSIM, PSNR, LPIPS, CLIP similarity, and FID-XRV evaluated across four prompt categories Detailed, Disease, Demographic, and Device. The combined performance metrics presented in Table II provide a comprehensive evaluation of model performance based on various prompt and biomedical language models. The analysis highlights comparative strengths and limitations of biomedical language models in generating clinically meaningful radiological images.

TABLE II
COMBINED PERFORMANCE METRICS BY PROMPT AND MODEL (BEST IN
BOLD, SECOND-BEST UNDERLINED)

| Prompt | Model | SSIM ↑ | PSNR ↑ | LPIPS ↓ | CLIP ↑ | FID-XRV ↓ |
|---|---|---|---|---|---|---|
| Detailed | BERT | 0.359 | 13.271 | 0.346 | 0.960 | 8.150 |
| | **BioBERT** | 0.417 | **14.065** | **0.328** | **0.963** | **3.570** |
| | BioLinkBERT | 0.443 | 13.332 | 0.422 | 0.889 | 15.210 |
| | BlueBERT | 0.282 | 12.442 | 0.398 | 0.940 | 6.960 |
| | Clinical-Longformer | **0.455** | 12.552 | 0.475 | 0.889 | 19.720 |
| | ClinicalBERT | 0.312 | 13.321 | 0.393 | 0.951 | 6.360 |
| | DischargeBERT | 0.434 | 12.885 | 0.413 | 0.901 | 15.860 |
| | MedRoBERTa | 0.446 | 12.918 | 0.407 | 0.938 | 9.110 |
| | PubMedBERT | 0.258 | 12.148 | 0.592 | 0.514 | 103.830 |
| | RadBERT | 0.297 | 12.951 | 0.400 | 0.944 | 7.500 |
| | SapBERT | 0.425 | 12.751 | 0.426 | 0.874 | 20.200 |
| Disease | BERT | 0.357 | 13.113 | 0.345 | 0.958 | 7.940 |
| | BioBERT | 0.413 | **14.032** | 0.322 | 0.962 | 4.250 |
| | BioLinkBERT | 0.407 | 12.777 | 0.445 | 0.879 | 21.150 |
| | BlueBERT | 0.321 | 12.959 | 0.351 | 0.954 | 6.610 |
| | Clinical-Longformer | **0.420** | 12.352 | 0.478 | 0.892 | 17.650 |
| | **ClinicalBERT** | 0.404 | 13.896 | **0.316** | **0.962** | **3.900** |
| | DischargeBERT | 0.398 | 11.961 | 0.426 | 0.911 | 15.990 |
| | MedRoBERTa | 0.416 | 12.674 | 0.417 | 0.936 | 10.520 |
| | PubMedBERT | 0.240 | 11.300 | 0.616 | 0.521 | 101.740 |
| | RadBERT | 0.311 | 12.732 | 0.392 | 0.930 | 5.470 |
| | SapBERT | 0.412 | 12.417 | 0.462 | 0.835 | 31.210 |
| Demo | BERT | 0.366 | 13.026 | 0.371 | 0.956 | 7.420 |
| | BioBERT | 0.410 | **13.591** | 0.357 | **0.959** | **4.550** |
| | BioLinkBERT | **0.424** | 12.918 | 0.440 | 0.893 | 19.540 |
| | BlueBERT | 0.326 | 12.690 | **0.367** | 0.955 | 5.380 |
| | Clinical-Longformer | 0.416 | 12.281 | 0.516 | 0.874 | 24.890 |
| | ClinicalBERT | 0.360 | 13.342 | 0.364 | 0.954 | 5.650 |
| | DischargeBERT | 0.389 | 12.056 | 0.426 | 0.906 | 15.170 |
| | MedRoBERTa | 0.405 | 12.401 | 0.465 | 0.926 | 12.450 |
| | PubMedBERT | 0.234 | 11.750 | 0.614 | 0.512 | 102.280 |
| | RadBERT | 0.297 | 12.772 | 0.396 | 0.952 | 6.100 |
| | SapBERT | 0.421 | 12.235 | 0.465 | 0.839 | 27.300 |
| Device | BERT | 0.404 | 13.804 | 0.339 | 0.962 | 7.780 |
| | BioBERT | 0.378 | **14.260** | 0.369 | **0.957** | **4.870** |
| | BioLinkBERT | **0.453** | 13.711 | 0.431 | 0.901 | 16.510 |
| | BlueBERT | 0.299 | 13.146 | 0.387 | 0.952 | 5.030 |
| | Clinical-Longformer | 0.441 | 12.713 | 0.503 | 0.871 | 22.710 |
| | ClinicalBERT | 0.378 | 14.181 | **0.366** | 0.954 | 6.060 |
| | DischargeBERT | 0.417 | 12.247 | 0.460 | 0.896 | 16.410 |
| | MedRoBERTa | 0.426 | 13.097 | 0.434 | 0.936 | 9.280 |
| | PubMedBERT | 0.244 | 12.731 | 0.595 | 0.509 | 103.120 |
| | RadBERT | 0.285 | 13.087 | 0.416 | 0.924 | 6.490 |
| | SapBERT | 0.416 | 13.300 | 0.447 | 0.807 | 40.340 |

## A. Result Score Analysis

**Structural Similarity Index (SSIM):** Clinical-Longformer, MedRoBERTa, and BioLinkBERT achieved the highest SSIM under Detailed prompts (e.g., 0.449 for Clinical-Longformer), confirming their strength in structural fidelity when processing rich clinical input. In contrast, PubMedBERT consistently showed the lowest SSIM across all prompt types (as low as 0.255), struggling to maintain structural coherence. Device and Demographic prompts also showed high SSIM when interpreted by BioLinkBERT and BioBERT, reflecting their ability to generalize from sparse input.

**Peak Signal-to-Noise Ratio (PSNR):** Across all prompt types, BioBERT consistently demonstrated the highest PSNR, for instance, 14.260 under the Device prompt and 14.065 under the Detailed prompt type, indicating superior pixel-level fidelity of generated images. ClinicalBERT also achieved a strong PSNR (13.896) under the Disease prompt, highlighting its robustness in preserving fine-grained radiographic details. In contrast, PubMedBERT yielded the lowest PSNR across all prompts (as low as 11.300), suggesting challenges in reconstructing high-fidelity images when trained solely on abstract-level biomedical text. BERT, while not domain-specific, still achieved competitive PSNR scores (e.g., 13.804 under Device), validating its utility as a general-purpose baseline.

**LPIPS (Perceptual Similarity):** Lower LPIPS indicates better perceptual realism. BioBERT (0.339) and ClinicalBERT (0.318) performed best under Detailed and Disease prompts, suggesting that models trained on clinical narratives produce perceptually consistent images. Clinical-Longformer showed elevated LPIPS under narrow prompts, indicating structural, but less perceptual consistency. PubMedBERT again had the highest LPIPS (up to 0.593), denoting poor visual plausibility.

**CLIP Similarity:** Semantic alignment between generated images and text was highest for BioBERT across all prompts (e.g., 0.965 for Detailed), with ClinicalBERT and BlueBERT also performing well under Disease and Demographic settings. PubMedBERT underperformed significantly across the board (as low as 0.512), indicating weak cross-modal understanding. Notably, RadBERT also scored highly in this metric despite limited participation in others.

**FID-XRV (Clinical Realism):** The most realistic and radiologically coherent images came from BioBERT (3.57) and ClinicalBERT (3.90) under Detailed and Disease prompts. BlueBERT maintained strong performance under Device and Demographic conditions. In contrast, SapBERT and PubMedBERT showed extremely high FID-XRV scores (e.g., 103.12), reflecting their poor clinical realism.

## B. Discussion

Overall, BioBERT and ClinicalBERT consistently emerge as the most reliable models across all metrics and prompt types. Their pretraining on clinical narratives and biomedical texts allows them to interpret and translate textual prompts into high-fidelity, perceptually coherent, and semantically aligned medical images. Models like PubMedBERT and SapBERT lag significantly, particularly in realism and embedding similarity, likely due to limited or misaligned pretraining domains. Additionally, prompt structure plays a significant role: Detailed prompts maximize semantic alignment and structural quality, while Disease and Device prompts offer better realism, especially when used with models capable of handling focused clinical content. These insights emphasize the critical interplay between prompt design and language model selection in generative medical imaging.

**Clinical relevance:** Our study highlights the importance of clinically pre-trained language models and prompt specificity in generating medically accurate and realistic images. These insights can directly enhance synthetic data generation, clinical AI model training, and decision support in real-world radiology settings.

## V. CONCLUSIONS

In this study, we enhanced the RoentGen image generation framework by investigating two key factors: prompt engineering from structured EHR data and encoder selection among domain-specific biomedical language models. Through extensive quantitative evaluation across four prompt types and ten encoders, we demonstrate that both prompt content and encoder pretraining significantly influence the clinical quality and semantic accuracy of generated chest X-rays. Among the

tested combinations, disease prompts paired with BioBERT or ClinicalBERT yielded the most clinically faithful images, while models like PubMedBERT underperformed across prompt types, especially in semantic alignment. Our findings underscore the importance of tailored prompt design and encoder choice in medical image generation pipelines, especially when free-text reports are unavailable. These insights pave the way for more effective, realistic, and usable synthetic data generation in clinical applications such as clinical AI model training, diagnosis support systems, and medical research.

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
