# OpenReview forum: "StructGen: Leveraging Structured EHR Prompts and Biomedical BERTs for Chest X-ray Synthesis"
_IEEE.org/EMBS/BHI/2025/Conference — BHI 2025_

### Official Review · Reviewer_TLab · 2025-07-15
**Generating synthetic Chest X-rays using RoentGen and encoders from EHR Data**

**Confidence:** 5
**Clarity Of Writing:** excellent
**Clinical Significance:** poor
**Methodological Novelty:** good
**Overall Rating:** 7

**Experiments And Results:**

great

**Questions For The Authors:**

Thank you for this paper on Generating synthetic Chest X-rays using RoentGen and encoders from EHR Data.
It is clear that having 4 types of prompts were used in experimental design. I am wondering if the clinical realism might be improved using available EHR clinical notes, because this is the place, all the important characteristics of the disease are written by physician and nurses.
Was there any attempt to compare the model generated image output with actual image output available (other than FID-XRV)?
Was there any attempt to verify the model generated image output by experienced radiologists?

**Strengths:**

Largest cohort size

**Summary Of The Paper:**

The paper compares synthetic chest X rays generated from RoentGen - latent diffusion-based image generation by using prompts and encoders. It is mentioned that detailed and disease prompts along with BIOBERT and ClinicalBERT achieved higher performance metrics such as SSIM, CLIP, LPIPS & FID-XRV.

**Weaknesses:**

Clinical actionability and real-world application.
I see, these models can be used for education purposes but not in clinical environment for prognosis and diagnosis.

---

### Official Review · Reviewer_E5ur · 2025-07-16
**StructGen: Leveraging Structured EHR Prompts and Biomedical BERTs for Chest X-ray Synthesis**

**Confidence:** 3
**Clarity Of Writing:** great
**Clinical Significance:** fair
**Methodological Novelty:** fair
**Overall Rating:** 4
**Final Rating:** 7

**Experiments And Results:**

fair

**Questions For The Authors:**

1) Why does the disease prompt strategy result in worse performance, especially when the disease is a critical component of an effective synthesized medical image?

**Strengths:**

1) The various prompt strategies, BERT encoders, and evaluation metrics used are well defined, and the authors do a great job differentiating the different components of their work.

2)The study is well structured, the dataset used is also well supported, and the analysis includes a comprehensive combination of variables (encoder, prompt strategy, eval metric).

**Summary Of The Paper:**

The authors test several BERT-based language models, guided by prompts generated from four EHR fields, to enhance the realism and alignment of synthesized images. The study utilizes a subset of the MIMIC-IV dataset containing chest X-rays, including both radiographs and their corresponding reports. The image-generating software used is RoentGen, an LDM that takes text embeddings and generates chest X-ray images. To evaluate the synthesized images, the authors employ four evaluation metrics and assess performance across four prompt strategies and ten BERT encoders. Among the ten BERT encoders, BioBERT and ClinicalBERT demonstrated the strongest performance. Each of the four prompt strategies showed favorable results depending on the evaluation metric used.

**Weaknesses:**

1) While I appreciate the comprehensive set of tools used, including encoders and prompt strategies, much of the content in Methodology Subsection C may be better presented in Table 1 for clarity. Additionally, evaluating ten different encoders seems excessive; it may be more effective to focus on a smaller set of state-of-the-art BERT encoders.

2) One of my main concerns with this evaluation is the poor performance of disease-specific prompts. This suggests that similarity scores may be driven more by attributes such as age, gender, or medical devices, which are important. However, in medical image synthesis, the focus is often on representing a specific condition. If the synthesized image does not clearly depict that condition, it is difficult to justify its similarity to the real image. Perhaps similarity could also be assessed using a disease recognition model or confirmed by a radiologist.

3) This is a minor point, but I would like to know whether the differences in performance between the BERT encoders and prompt strategies are statistically significant and meaningful, or if they might be due to random variation.

---

### Official Review · Reviewer_Hdeo · 2025-07-16
**StructGen: Leveraging Structured EHR Prompts and Biomedical BERTs for Chest X-ray Synthesis**

**Confidence:** 4
**Clarity Of Writing:** good
**Clinical Significance:** great
**Methodological Novelty:** good
**Overall Rating:** 8

**Experiments And Results:**

great

**Questions For The Authors:**

Points for improvement in the document:
-	It is common for articles not to include references in the abstract.
-	Acronyms are defined the first time they appear in the text (for example, BERT in the abstract section, CNN, …)
-	StructGen: this term should appear in the text, not only in the title and legend of figure 1.

**Strengths:**

Key strengths of this work include the comprehensive state-of-the-art overview and the thorough experimental setup, which evaluates four distinct prompt types and ten specialized biomedical language model encoders. This extensive analysis allows for a deeper understanding of how prompt design and encoder choice impact the clinical relevance and semantic fidelity of generated chest X-rays.

**Summary Of The Paper:**

This study enhances the RoentGen image generation framework by evaluating how structured prompts from electronic health records and different biomedical language model encoders influence the quality of synthetic chest X-rays. Through testing four prompt types and ten encoders, it is found that both the prompt content and the encoder's pretraining strongly affect the clinical realism and semantic accuracy of the images. The best results were achieved using disease-specific prompts with BioBERT or ClinicalBERT, while models like PubMedBERT showed weaker performance, especially in maintaining semantic consistency.

**Weaknesses:**

No issues to report.

---

### Official Review · Reviewer_aQjf · 2025-07-16
**Text encoder benchmark and prompt engineering for Chest X-ray synthesis**

**Confidence:** 5
**Clarity Of Writing:** great
**Clinical Significance:** fair
**Methodological Novelty:** fair
**Overall Rating:** 7

**Experiments And Results:**

great

**Questions For The Authors:**

- How does a non-domain specific language model (e.g., BERT) would perform in this case? Given the level of overlap between the training data among existing clinical text encoder, I think it is critical to include plain BERT as a baseline.
- I am also interested in a comparison of PSNR
- If the evaluation code is planned to be open sourced, that should be included in the paper.

**Strengths:**

- comprehensive benchmark of existing clinical text encoders
- good choice of prompt engineering strategies

**Summary Of The Paper:**

This paper presented how structured EHR prompts and language model pertained on biomedical data affect chest X-ray synthesis using latent diffusion model. Four EHR-derived prompt strategies are evaluated including detailed, disease, demographic, and device paired with ten domain specific text encoder. Results show that BioBERT and ClinicalBERT consistently produced competitive results when combined with disease or detailed prompting.

**Weaknesses:**

- discussion of the results should benefit from the fact that clinical-specific text encoders differ significantly in their training corpora, domain focus, and design objectives
- the paper does not present how well generated images integrate with downstream tasks, thus lack clinical significance
- only a single architecture is used without exploring the observed trends generalize across other generative architectures